#### of Photosynthesis-irradiance parameters marine 1 phytoplankton: synthesis of a global data set 2

Heather A. Bouman<sup>1</sup>, Trevor Platt<sup>2</sup>, Martina Doblin<sup>3</sup>, Francisco G. Figueiras<sup>4</sup>, Kristinn Gudmundsson<sup>5</sup>, Hafsteinn G. Gudfinnsson<sup>5</sup>, Bangqin Huang<sup>6</sup>, Anna Hickman<sup>7</sup>, Michael Hiscock<sup>8</sup>, Thomas Jackson<sup>2</sup>, Vivian A. Lutz<sup>9</sup>, Frédéric Mélin<sup>10</sup>, Francisco Rey<sup>11</sup>, Pierre Pepin<sup>12</sup>, Valeria Segura<sup>9</sup>, Gavin H. Tilstone<sup>2</sup>, Virginie van Dongen-Vogels<sup>3</sup>, Shubha 3 4 5

6 Sathyendranath<sup>2</sup> 7

- 8 <sup>1</sup>Department of Earth Sciences, University of Oxford, Oxford, OX1 3AN, United Kingdom
- 9 <sup>2</sup>Plymouth Marine Laboratory, Prospect Place, The Hoe, PL1 3DH, United Kingdom
- <sup>3</sup> Plant Functional Biology and Climate Change Cluster, Faculty of Science, University of Technology Sydney, PO 10
- 11 Box 123 Broadway, Sydney, NSW 2007, Australia
- 12 Instituto de Investigaciones Marinas (CSIC), Eduardo Cabello 6, 36208 Vigo, Spain
- 13 Marine Research Institute, PO Box 1390, 121 Reykjavík, Iceland
- <sup>6</sup> State Key Laboratory of Marine Environmental Science/Key Laboratory of Coastal and Wetland Ecosystems, 14 15 Ministry of Education, Xiamen University, Xiamen, Fujian 361005, China
- 16 Ocean and Earth Science, University of Southampton, National Oceanography Centre Southampton, European 17 Way, Southampton, SO14 3ZH, United Kingdom
- 18 <sup>8</sup> United States Environmental Protection Agency, Ariel Rios Building, 1200 Pennsylvania Avenue, Washington 19 D.C., 20460, U.S.A.
- <sup>9</sup> Instituto Nacional de Investigacion y Desarrollo Pesquero, Mar del Plata, Argentina
- <sup>10</sup> Institute for Environment and Sustainability, Joint Research Centre, European Commission, Ispra 21027, Italy
- <sup>11</sup> Institute of Marine Research, c/o Department of Biological Sciences, University of Oslo, POB 1066, N-0316, Oslo, Norway
- 20 21 22 23 24 25 26 27 <sup>12</sup> Fisheries and Oceans Canada, Northwest Atlantic Fisheries Centre, PO Box 5667, St John's, Newfoundland, A1C 5X1, Canada
- - Correspondence to: Heather A. Bouman (Heather.Bouman@earth.ox.ac.uk)
- Abstract. The photosynthetic performance of marine phytoplankton varies in response to a
- variety of factors, environmental and taxonomic. One of the aims of the MArine primary
- Production: model Parameters from Space (MAPPS) project of the European Space Agency is
- to assemble a global database of photosynthesis-irradiance (P-E) parameters from a range of
- oceanographic regimes as an aid to examining the basin-scale variability in the
- photophysiological response of marine phytoplankton and to use this information to improve
- the assignment of *P*-*E* parameters in the estimation of global marine primary production using 35
- satellite data. The MAPPS *P-E* Database, which consists of over 5000 *P-E* experiments,
- provides information on the spatio-temporal variability in the two P-E parameters (the
- assimilation number,  $P_m^B$ , and the initial slope,  $\alpha^B$ , where the superscripts B indicate 38
- normalisation to concentration of chlorophyll) that are fundamental inputs for models (satellite-39
- based and otherwise) of marine primary production that use chlorophyll as the state variable.
- Quality-control measures consisted of removing samples with abnormally-high parameter

Science Scienc

- values and flags were added to denote whether the spectral quality of the incubator lamp was
- used to calculate a broad-band value of  $\alpha^{B}$ . The MAPPS database provides a
- photophysiological dataset that is unprecedented in number of observations and in spatial
- coverage. The database would be useful to a variety of research communities, including marine
- ecologists, biogeochemical modellers, remote-sensing scientists and algal physiologists. The
- compiled data are available at doi:10.1594/PANGAEA.874087 (Bouman et al., 2017).

#### 7 1 Introduction

9 Although global estimates of marine primary production tend to converge on a number around 10 40-50 GT per annum, the accuracy and precision on regional scales of the estimation 11 protocols remain relatively poor, partly as a result of an incomplete understanding of how the 12 photosynthetic performance of marine phytoplankton varies in the global ocean (Carr et al. 13 2006, Lee et al. 2015). Photosynthesis-irradiance (P-E) parameters derived from carbon uptake 14 experiments conducted over a controlled range of available-light levels provide a means of 15 comparing the photosynthetic characteristics of marine phytoplankton across different natural 16 populations and cultured isolates (Platt and Jassby 1976, Prézelin et al. 1989, MacIntyre et al. 17 2002). The P-E experiment exposes algal cells to a range of light intensities from near-zero to 18 those levels typically available at the sea surface (Lewis and Smith 1983, Babin et al. 1994). 19 The photosynthetic rates are then normalised to the concentration of chlorophyll-a (a useful and 20 practical index of phytoplankon biomass relevant for photosynthesis) found within the sample. 21 This normalisation serves two purposes: first, dividing by pigment biomass reduces the 22 variability of photosynthesis rates due to differences in biomass alone, facilitating the 23 comparison of photosynthetic performance across trophic gradients, and second, chlorophyll-24 normalised photophysiological parameters may be applied in the estimation of primary 25 production over large scales by using satellite-derived maps of chlorophyll concentration 26 (Longhurst et al. 1995, Antoine and Morel 1996). A schematic diagram showing the biomass-27 normalised data generated from these experiments plotted against the light intensity at which 28 each bottle was incubated is shown in Fig. 1 to illustrate how the ensemble of data, when fitted 29 to a suitable non-linear equation, forms a *P*-*E* curve. The curve may be represented by a variety 30 of mathematical forms (Jassby and Platt 1976, Platt et al. 1980). In cases where 31 photoinhibition is negligible, all equations suitable for describing the P-E curve can be

- represented using just two parameters: the initial slope,  $\alpha^{B}$ , which represents the photosynthetic
- efficiency under light levels close to zero, and the asymptote of the curve,  $P_m^B$ , which is the
- photosynthetic rate at light saturation (Jassby and Platt 1976, Platt et al. 1980, Sakshaug et al.
- 1997).
- 2 Data
- Chorophyll-a concentrations and photosynthesis-irradiance (P-E) parameters collected from
- four oceanic domains and 35 biogeochemical provinces (Longhurst 2007, Table 1) were
- compiled from individual investigators and online data repositories (Table 2). *P-E* data were
- obtained by <sup>14</sup>C and <sup>13</sup>C (Argentine Sea) uptake experiments, with incubation times varying
- from 1.5 to 4 hours. Chlorophyll concentrations used to normalise the carbon fixation rates
- were measured using either High Performance Liquid Chromatography or the standard
- fluorometric method (Mantoura et al. 1997). The environmental variables and photosynthetic
- parameters included the MAPPS database and their corresponding units are listed in Table 3.
- Further details on the experimental methodology for individual field campaigns are provided in
- the original publications (see Table 2).
- Table 2 includes information on which functional form was fitted to the *P*-*E* data for each of the datasets used in this study. In cases where photoinhibition was absent (photosynthetic rates stayed independent of irradiance in the light-saturated range), or where the fit was applied to
- 22 data unaffected by photoinhibition, a two-parameter curve fit was used, of the form:

$$P^{\rm B}(E) = P_m^{\rm B} \tanh\left(\frac{\alpha^{\rm B} E}{P_m^{\rm B}}\right),\tag{1}$$

23

where  $P^{B}(E)$  is the chlorophyll-normalised photosynthetic rate (mg C (mg chl-a)<sup>-1</sup> h<sup>-1</sup>) and *E* is the available light, which in this study is expressed in µmol quanta m<sup>-2</sup> s<sup>-1</sup>. The light saturation parameter,  $E_{k}$ , is defined by the following relationship

$$E_k = \frac{P_m^B}{\alpha^B} \tag{2}$$

- and is illustrated in Figure 1 by the drawing a line from the intersection of the initial slope withthe plateau of the curve onto the abscissa and has dimensions of irradiance.
- 30

- 1 In most cases, however, data were fit to the three-parameter function of Platt et al. (1980),
- 2 which describes also the decrease in photosynthetic rate with irradiances much higher than
- 3 saturating light levels, as follows:

4

$$P^{B}(E) = P_{s}^{B}\left(1 - \exp\left(-\frac{\alpha^{B}E}{P_{s}^{B}}\right)\right) \exp\left(-\frac{\beta^{B}E}{P_{s}^{B}}\right),$$
(3)

- 5 6
- 7 where  $\beta^{B}$  is the photoinhibition parameter describing the decrease in photosynthetic rate at high
- 8 irradiance and  $P_s^B$  is the hypothetical maximum photosynthetic rate in the absence of
- 9 photoinhibition. Hence when  $\beta^B = 0$ ,  $P_s^B = P_m^B$ . When photoinhibition was present, values of
- 10  $P_m^B$  were derived using the following equation:

$$P_m^B = P_s^B \left(\frac{\alpha^B}{\alpha^B + \beta^B}\right) \left(\frac{\beta^B}{\alpha^B + \beta^B}\right)^{\frac{B}{\alpha}}.$$
(4)

11

#### 12 Quality Control for the MAPPS PE database

#### 13 **3.1 Experimental conditions**

The *P-E* experiments were performed in incubators that maintained samples under in situ

- temperature conditions using either temperature-controlled water baths or the ship's underway
- water supply. Samples where incubation temperatures differed from in situ temperatures by
- more than 2 °C were removed from the database. It is well known that the light spectrum has a
- significant effect on the magnitude of light-limited photosynthesis ( $\alpha^{B}$ ) and the derived light
- saturation parameter ( $E_k$ ) (Kyewalyanga et al. 1997; Schofield et al. 1991). We have included
- in the database quality flags indicating whether a correction factor for the spectrum of the lamp
- was applied to obtain a readily-intercomparable broad-band (white light) value (e.g.
- Kyewalyanga et al. 1997, Xie et al. 2015). This broad-band  $\alpha^{B}$  combined with information on
- the shape of the phytoplankton absorption spectrum has been shown to provide an accurate
- estimate of the photosynthetic action spectrum  $\alpha^{B}(\lambda)$ . The correction factor X can be used to
- convert the measured  $\alpha^{B}$  from the incubation experiment using a given artificial light source to
- an estimate of  $\alpha^{B}$  if the sample were subject to a spectrally-neutral light environment: it is the

- 1 ratio of the unweighted mean absorption coefficient of phytoplankton  $(\bar{a}_p)$  to the mean
- 2 absorption coefficient weighted by the shape of the emission spectrum of the lamp source  $(\bar{a}_L)$

$$X = \frac{\bar{a}_p}{\bar{a}_T},\tag{5}$$

4 where  $\bar{a}_p$  is determined by

$$\bar{a}_p = \frac{\int_{400}^{700} a_p(\lambda) d\lambda}{\int_{400}^{700} d\lambda},$$
(6)

5 and  $\overline{a}_T$  is computed as 7

 $\bar{a}_T = \frac{\int_{400}^{700} a_p(\lambda) E_T(\lambda) d\lambda}{\int_{400}^{700} E_T(\lambda) d\lambda}.$ (7)

8

3

9 The incubators in this study used a range of light sources, including tungsten halogen, halogen, 10 metal halide, and fluorescent lamps. Tungsten halogen lamps are the most commonly-used 11 light source in *P-E* experiments because they provide intensities sufficiently high to resemble 12 irradiances at the sea surface (~ 2000  $\mu$ mol quanta m<sup>-2</sup> s<sup>-1</sup>). One limitation of using tungsten 13 lamps is that they have a spectrum heavily weighted towards the red and infrared (see Fig. 2), 14 unless the light first passes through a filter that removes the red emission. Table 4 describes the

15 various lamps and filters used in the *P*-*E* incubators used in this study.

16 To estimate the impact of a tungsten halogen lamp compared with a white light source on the magnitude of the  $\alpha^{B}$  and consequently  $E_{k}$ , which is derived from estimates of  $\alpha^{B}$  (Equation 2), 17 we used a dataset from the North Atlantic that spanned several decades. From 1994, P-E data 18 19 have been corrected for the spectrum of the lamp source following the method of Kyewalyanga et al. (1997), whereas prior to 1994, no correction was made due to a lack of information on the 20 21 excitation spectrum of the lamp and the absorptive properties of the phytoplankton 22 communities. By comparing data from similar regions and seasons as lamp sources have 23 changed, we are able to assess how the light source may cause variability in the photosynthetic parameter  $\alpha^{B}$ . In the post-1994 dataset, with corresponding lamp and absorption spectra, the 24 25 correction factor X varied from 1.30 to 2.06 (mean=1.70 with a standard deviation of 0.15). This variation in X (Fig. 3a) is sufficient to account for difference in magnitudes of  $\alpha^{B}$  obtained 26 27 using incubators with different light sources across the North Atlantic cruise dataset (Fig. 3b). Note that potential errors in the computation of primary production due to changes in  $\alpha^{B}$  caused 28 by spectral differences in light sources will be most acute deeper in the water column, where 29

- the influence of the magnitude of  $\alpha^{B}$  on primary production is greatest (Ulloa et al. 1997,
- Bouman et al. 2000a) and thus errors for integrated water-column primary production will be
- modest since productivity rates are highest at the surface and decrease in an exponential
- manner once  $E(z) \leq E_k$  (Ulloa et al. 1997, Bouman et al. 2000a).

#### 5 3.1.1 Theoretical maxima

- The photophysiological constraints of marine phytoplankton are well known, and provide a
- useful check on the quality of the carbon-uptake experiments. The theoretical maximum
- quantum yield of carbon fixation ( $\phi_m^T$ ) is 0.125 mol C (mol quanta)<sup>-1</sup> (Platt and Jassy 1976,
- Sakshaug et al. 1997). The realised maximum quantum yield of photosynthesis ( $\phi_m$ ) is
- calculated by dividing  $\alpha^{B}$  by  $\overline{a^{*}}$ , the chlorophyll-specific absorption coefficient of
- phytoplankton averaged over the visible spectrum (Platt and Jassby 1976) and multiplying by a
- factor of 0.0231, which converts milligrams to moles of C, µmols to moles of photons, and
- 13 hours to seconds. Values of  $\phi_m$  were calculated using either simultaneous measurements of  $\overline{a^*}$ ,
- or estimates derived from a global relationship between chlorophyll concentrations and  $\overline{a^*}$
- (Bouman, unpublished data) and samples with  $\phi_m$  well above the theoretical maximum (>0.15
- 16 mol C (mol quanta)<sup>-1</sup>) were discarded from the database. We also set a lower limit for the light
- saturation parameter  $P_m^B$  of 0.2 mg C (mg chl-a)<sup>-1</sup> h<sup>-1</sup> and the initial slope  $\alpha^B$  of 0.002 mg C (mg
- $chl-a)^{-1}h^{-1}$  (µmol quanta m<sup>-2</sup> s<sup>-1</sup>)<sup>-1</sup>. Data from experiments on sea-ice algae with chl-a
- concentrations exceeding 50 mg chl-a  $m^{-3}$  were also removed. Using these criteria, 278
- experiments were excluded from the global database.

#### 21 4 Results

#### 22 4.1 Spatio-temporal patterns of the MAPPS P-E database

In this study we adopt the Longhurst's (2007) geographical classification system of domains

and provinces to partition the global dataset according to the prevailing physical conditions that

- shape the structure and function of phytoplankton communities over large (basin) scales. The
- rationale behind using Longhurst's approach to estimate primary productivity is that physical
- forcing dictates the supply of nutrients and the average irradiance within the surface mixed
- layer and these factors directly impact the physiological capacity of algal cells. The four

1 domains (also referred to as Longhurst biomes) are found in each ocean basin and are subject to 2 distinct mechanisms of physical forcing: in the polar domain the density structure of the surface 3 layer is strongly influenced by sea ice melt; in the westerlies domain the mixed layer dynamics 4 are governed by a local balance of heat-driven stratification and wind-driven turbulent mixing 5 by winds; in the trades domain, the depth of the mixed layer is governed by geostrophic 6 responses to seasonal changes in the strength and location of the trade winds and in the coastal 7 domain, terrestrial influx of freshwater and interaction of local winds with topography play a 8 critical role in governing ecosystem properties. The next level of partition is biogeochemical 9 provinces, which embrace a wider set of environmental factors that govern regional ocean 10 circulation and stratification that in turn influence ecological structure. Although it would be 11 preferable that both domain and provincial boundaries were dynamic to accommodate seasonal, 12 annual and decadal changes in ocean circulation (Devred et al. 2007), to exploit the entire 13 MAPPS P-E parameter dataset, which contains a large number samples that were collected 14 prior to the launch of ocean-colour satellites, we used the fixed rectilinear boundaries of 15 Longhurst (2007) with the understanding that some of the within-province variability may be the result of the estimated and actual provincial boundaries being spatially offset. 16 17 18 Roughly half of the quality-controlled samples were collected from within the upper 20 m of 19 the water column, accounting for approximately 54% of the dataset. Most of the data fall 20 within the Atlantic Basin (Fig. 4), with a large region of the Pacific Basin being grossly 21 undersampled in both space and time. The latitudinal coverage of the database is relatively 22 sparse in the tropics and the mid-latitudes of the Southern Hemisphere. The seasonal 23 distribution of data shows the majority of samples were collected during the spring (40%). 24 summer (34%) and fall (22%), and only 3% of the samples collected during the winter period (Table 2, Fig. 5). Across all seasons, the dataset covers a range of trophic conditions, from 25 highly oligotrophic conditions (0.02 mg m<sup>-3</sup>) to spring bloom conditions (39.8 mg m<sup>-3</sup>) (Fig. 6). 26 The dynamic range of the photosynthetic parameters was similar to that reported in other global 27 studies, with values  $P_m^B$  ranging from 0.21 and 25.91 mg C (mg chl-a)<sup>-1</sup> h<sup>-1</sup> with an average 28 value of 3.11 and a standard deviation = 2.28 and values of  $\alpha^{B}$  ranging from 0.002 to 0.373 mg 29 C (mg chl-a)<sup>-1</sup> h<sup>-1</sup> (umol quanta m<sup>-2</sup> s<sup>-1</sup>)<sup>-1</sup> with an average value of 0.043 and a standard 30 deviation of 0.034. 31 32

1 The *P*-*E* parameters exhibited both spatial (between provinces) and temporal (between seasons) 2 differences. In general, values of the assimilation number increased with decreasing latitude 3 (Table 2) and tended to be higher during the summer months in temperate marine systems. 4 However, the seasonal and latitudinal bias in data coverage has important implications for 5 variability in parameter values in the dataset because of the environmental conditions known to 6 influence phytoplankton photophysiology. High-latitude samples will be associated with lower 7 temperatures, which may limit their maximum photosynthetic rate for carbon fixation (Smith 8 Jr. and Donaldson et al. 2015), and this is reflected in the generally low values of  $P_m^B$  in the 9 Boreal (BPLR) and Austral (APLR) Polar provinces. Geographical variation in surface 10 irradiance may also explain the lower values of  $P_m^B$  in high latitudes compared with low latitudes. The combination of lower surface irradiances and deep convective mixing in high 11 12 latitudes results in markedly lower light levels within the mixed-layer, which may result in 13 photoacclimation to lower light levels, by modulating pigment content per cell and hence the 14 carbon-to-chlorophyll ratio (Cullen et al. 1982, Sathyendranath et al. 2009). However, it is 15 important to note that some of the polar samples were collected in regions highly influenced by 16 sea-ice melt, which may lead to the formation of a fresh, shallow and highly-stable mixed layer, 17 and consequent higher average light level than would be the case for deeper mixing. 18 19 The paucity of winter data reduces the number of samples with cells acclimated to low growth 20 irradiances. The number of observations is also low within the tropical and sub-tropical oceans 21 which are characterised by warm and hence strongly-stratified mixed layers. Phytoplankton cells in these regions tend to have higher upper bounds of  $P_m^B$ , due to the combined effect of 22 23 warmer sea-surface temperatures and the acclimation to high-light conditions. Such spatio-24 temporal patterns in the *P*-*E* parameters are likely driven by changes in oceanographic 25 conditions (temperature, stratification, macro- and micronutrient availability) (Geider et al. 26 1996) as well as in community structure and other biotic processes that may consume cellular 27 energy at the expense of carbon fixation (Puxty et al., 2016).

28

#### 29 4.2 Relationship between the maximum photosynthetic rate and the initial slope

# Science Scienc

- Strong correlations have been reported between the two *P-E* parameters  $P_m^B$  and  $\alpha^B$ , which have been explained on both ecological and photophysiological grounds (Platt and Jassby 1976, Côté and Platt, 1983, Behrenfeld et al. 2004). The MAPPS dataset shows that the data fall largely within the bounds of  $E_k$  values between 20 and 300 µmol quanta m<sup>-2</sup> s<sup>-1</sup> (Fig. 7). In general, high-latitude samples (> 65°) tend to have lower  $P_m^B$  values for a given value of  $\alpha^B$  ( $E_k$  values averaging 57.7 µmol quanta m<sup>-2</sup> s<sup>-1</sup> with 10.0% of the data falling above 100 µmol quanta m<sup>-2</sup> s<sup>-1</sup>
  - <sup>1</sup>) when compared with low latitude samples between 40° N and 40° S ( $E_k$  values averaging
  - 152.4  $\mu$ mol quanta m<sup>-2</sup> s<sup>-1</sup>, with 57.1% of the values falling above 100  $\mu$ mol quanta m<sup>-2</sup> s<sup>-1</sup>).
- When  $E_k$ , is plotted as a function of latitude (Fig. 8) for open-ocean samples within the top 25m
- of the water column, a clear pattern emerges, with higher latitude samples being characterised
- by lower values, whereas data from the mid-to-low latitudes had, on average, higher values,
- although considerable scatter was observed over the entire range of temperatures. To illustrate
- the depth-dependent change in  $E_k$  due to vertical changes in irradiance, data from cruises that
- predominantly sampled stratified, oligotrophic regions (DCM and AMT cruises) are plotted
- against the sample depth. The strong depth-dependence of the photoacclimation parameter is
- consistent with other open-ocean studies (Babin et al. 1996). The latitudinal and depth
- dependence of  $E_k$  was also reported in a study which used a subset (N=1862) of the MAPPS
- database from the North Atlantic spanning the Tropics to the Arctic: 55% of the variance in  $E_k$
- could be explained using depth, latitude, temperature, nitrate and surface noon irradiance as
- predictive variables (Platt and Sathyendranath 1995).

#### 22 5 Discussion

- Predicting the photosynthetic efficiency of phytoplankton cells remains one of the major challenges in determining marine primary production using remote sensing data (Carr et al. 2006). The MAPPS database of *P-E* parameters allows us to assess the global variability in phytoplankton photophysiological parameters and could be used to validate models that aim to provide a mechanistic understanding of changes in the photosynthetic parameters. Here, we attempt to explain the spatial patterns in the dataset drawing on our current understanding of the key environmental factors governing variability in both  $P_m^B$  and  $\alpha^B$ .

- A positive correlation between  $P_m^B$  and  $\alpha^B$  has been attributed to a variety of physiological and
- ecological factors, including changes in the allocation of ATP and NADPH to carbon fixation
- (Behrenfeld et al., 2004), as well as changes in phytoplankton community structure (Côté and
- Platt, 1983). To disentangle the ecological from the physiological sources of variability is not
- straightforward, unless additional information on the taxonomic composition and
- photoacclimatory status of natural phytoplankton samples is available. Moreover, culture
- studies have invoked viral infection as another potential source of variability that is poorly
- understood in natural marine systems (Puxty et al. 2016).
- Both taxon-specific and size-specific differences in  $P_m^B$  and  $\alpha^B$  have been reported in both
- culture and field studies (Bouman et al. 2005, Côté and Platt 1983, 1984, Huot et al. 2013, Xie
- et al. 2015). As new remote-sensing algorithms are now starting to yield information on the
- size and taxonomic structure of phytoplankton, it would be useful to derive additional
- information on the *P*-*E* response of key phytoplankton taxa and size classes, especially those
- implicated as playing key roles in ocean biogeochemical cycles (LeQuéré et al. 2005, Nair et al.
- 2008, Bracher et al. 2017). Although detailed information on the taxonomic and size structure
- of ship-based experiments was lacking for several of the samples included in this dataset, more
- recent studies include some measure of phytoplankton community structure, whether it be from
- use of pigment markers, size fraction of pigment and/or productivity, or cell counts. Although
- there is a question as to what the standard indices of community structure should be that can
- help account for community-based variation in the photophysiological parameters across
- datasets, it is likely that information on gross community structure alone will not account for a
- large fraction of the variability in *P*-*E* parameters, especially across regions or seasons with
- different environmental forcing (Bouman et al. 2005, Smith Jr. and Donaldson 2015) or
- resident ecotypes (Geider and Osborne 1991).

The high range of photosynthetic parameters recorded at lower latitudes is largely caused by

- depth-dependent changes due to photoacclimation and photoadaptation (Babin et al. 1996,
- Bouman et al. 2000b, Huot et al. 2007) in highly stratified waters. Strong vertical gradients in
- nutrient supply and growth irradiance lead to a vertical layering of ecological niches resulting
- in strong vertical gradients in species composition and in the case of marine picocyanobacteria,
- high-light and low-light ecotypes, are observed (Johnson et al. 2006, Zwirglmaier et al. 2007).

| 1  | Although depth-dependent variability in the photosynthetic parameters can be examined in the          |
|----|-------------------------------------------------------------------------------------------------------|
| 2  | MAPPS dataset, in particular $E_k$ (Fig. 9), it has been argued that optical depth may be a more      |
| 3  | useful predictor of changes in the P-E parameters resulting from vertical changes in the              |
| 4  | photoacclimatory status of phytoplankton cells (Babin et al. 1996, Bouman et al. 2000b). In           |
| 5  | highly stratified and stable seas such as the oligotrophic gyres this may be the case, yet in more    |
| 6  | dynamic ocean conditions such as the Beaufort Sea, optical depth has been shown to have no            |
| 7  | more predictive skill, and sometimes less, than using depth alone (Huot et al. 2013). It is           |
| 8  | important to note that diel changes in the P-E parameters were not taken into account in this         |
| 9  | meta-analyses due to a lack of information on the time of sample collection in a significant          |
| 10 | number of observations, which can be a significant source of variability (MacCaull and Platt          |
| 11 | 1977, Prézelin and Sweeney 1977, Prézelin et al. 1986, Harding et al. 1983, Cullen et al. 1992,       |
| 12 | Bruyant et al. 2005). However, as noted in the study of Babin and co-authors (1996) such diel         |
| 13 | and day-to-day variability in the $P$ - $E$ parameters is likely to be far smaller when compared with |
| 14 | differences across biogeochemical provinces subject to markedly different environmental               |
| 15 | forcing.                                                                                              |
| 16 |                                                                                                       |
| 17 | Clear latitudinal differences in the range of $E_k$ values are revealed in the MAPPS dataset (Fig.    |

8) which suggests that  $E_k$  may be controlled by environmental factors that vary strongly with

latitude, such as temperature and the availability of light. Figure 8 shows that samples collected

from high-latitude environments, such as the Laborador Sea, the (sub)Arctic and the Southern

Ocean have markedly lower  $E_k$  values, reflecting the physical constraints of low temperatures

and, in some cases, low light levels (Harrison and Platt 1985). The physical dynamics of the

upper ocean and their impact on temperature and light conditions have been shown to play a

dominant role in governing the photosynthetic performance of polar and temperate marine

phytoplankton (Harrison and Platt, 1986, Bouman et al, 2005).

#### 26 6 Recommendations for use

The MAPPS database of photosynthesis-irradiance parameters may be used to compute marine

primary production using remotely-sensed data on ocean colour (e.g. Longhurst et al. 1995).

The data present in Table 2 provide seasonal values of the photosynthetic parameters for

Longhurst biogeochemical provinces, whose geographical boundaries can be found online at

http://www.marineregions.org/. This dataset will also serve as a useful resource for validation

- of other methods used to extrapolate *P*-*E* parameters to basin scales using ocean observables
- (Antoine and Morel 1996, Platt et al. 2008, Saux-Picart et al. 2014). The MAPPS database will
- allow such extrapolation procedures to be further tested and refined over a much larger range of
- geographical and biogeochemical domains or for new approaches of parameter assignment to
- be developed.

#### 7 6 Data availability

- The compiled data set containing 5711 individual photosynthesis-irradiance experiments is
- and corresponding metadata are available on PANGAEA (<u>https://doi.pangaea.de</u>): doi:
- 10.1594/PANGAEA.874087 (Bouman et al. 2017).

#### 11 Acknowledgements

- The authors wish to thank the research scientists, technicians, students and crew who
- contributed to the collection of these data. In particular, we acknowledge the significant
- contributions made by Brian Irwin, Jeff Anning, Gary Maillet, Christine Hanson, Brian
- Griffiths, James McLaughlin, Richard Matear, and Andy Steven.

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
