# Peer review of "of Photosynthesis-irradiance parameters marine 1 phytoplankton: synthesis of a global data set 2"

_Earth System Science Data, 2017_

## Referee Comment (RC1) · Anonymous Referee #1 · 20 Jun 2017

The manuscript of Bouman et al. synthesizes a global data set on P-E parameters, and tries to insure equality among all experiments to provide a data set that is truly comparable. The data set is good, and while some measurements that I am familiar with are not included, I suspect that this is true for most reviewers, even those that have published on the topic. I am not sure how ESSD treats the addition of new data, but no matter – that is not an issue here.

The analysis is solid; no problems. I would have liked to see some discussion about the variations among selected geographic regions or bio-provinces, but understand that the static description of those regions makes it difficult. Is it possible to run simple

statistics to look for significant spatial differences? I admit I am not sure how powerful they would be, but is the reader to conclude that differences among the broad regions are insignificant? Perhaps a more strongly worded conclusion on that result would be helpful.

The authors also discuss the effects of phytoplankton composition on the parameter estimates. I wonder if using a broad description of pigments as available in ESSD might provide some insights into the variations observed in P-E parameters?

In short, the paper is well written and a useful composite of P-E data available. I recommend publication with minor changes.

---

## Short Comment (SC1) · 6 Jul 2017

This study offers a comprehensive and unprecedented global database of photosynthesis-irradiance parameters derived from more than 5000 experiments. The database encompasses quality-controlled parameters of different oceanographic regimes providing a global scale insight of phytoplankton photophysiological variation. In my opinion, it is very well written and illustrated and it is ready for publication.

I would suggest a consistence in tables' format. Also, in pag 8, line 3, I believe the sentence refers to table 5 instead of table 2 and in page 9 (line 14-16) you could refer to Figure 9.

---

## Referee Comment (RC2) · A. J. Poulton (Referee) · 11 Aug 2017

The manuscript presents a unique and valuable dataset of photosynthetic parameters of marine algal with a global perspective. Such a dataset is extremely valuable to global attempts to measure (satellite-derived) and model marine primary production and I can see this dataset being widely used. I have mainly minor comments and suggestions.

P1, Ln 39: Concentration of chlorophyll – please specify chlorophyll-a, or are some of the normalisation against total chlorophyll? I assume the former is relevant based on reading the introduction and methods. As chlorophyll-a from both fluorometric and HPLC measurements are used for normalisation, should this not be another flag or

comment in the database? Alternatively, the authors should add some comment on the sensitivity of derived parameters from fluorometric versus HPLC measures of chlorophyll biomass.

P2, Ln 4: Replace 'would' with 'will'.

P6, Ln 18-19: More rationale is needed to explain why the data from sea-ice algae with chlorophyll-a concentrations greater than 50 mg chl-a m-2 were removed. I can see why but a report detailing the construction of a database needs to establish clear rationale for data-removal (especially given that such a database may be expanded in the future).

P6, Ln 23 onwards: Here it would be good to make clear that the application of the Longhurst (1995, 1998) provinces is the author's choice to represent the eco-geographical spread of the data and is not inherent in the database.

P7, Ln 25-26: Please make it clear that chlorophyll-a concentrations have been used to classify trophic conditions (i.e. it is unclear (to the uninitiated) what 0.02 mg m-3 and 39.8 mg m-3 refer to, and there are other index's that could be used to examine trophic conditions).

P7, Ln 28-29: How are the data distributed? Have the authors considered using a geometric mean rather than an arithmetic mean? Rates of photosynthesis and chlorophyll-a concentrations range from very low values (e.g., oligotrophic waters) up to extremely high values (e.g., ice-edge bloom). Consequently, photosynthetic rates (and derived parameters) can vary over several orders of magnitude, and appear to exhibit a log-normal distribution (e.g. Fig 6). In this case maybe a geometric mean, rather than arithmetic mean, may better represent the data?

P11, Ln 18-19: What about nutrient availability as an environmental factor which varies strongly with latitude?

Figs. 3-8: Please ensure that the axis labels and data-points are clear in the final

publication sized version of the manuscript to ensure they can be clearly read (some of the early figures have slightly small text for the axis).

Fig. 8 and 9: The density (or heat plot) component of the plots is not mentioned in the figure legends.

---

## Author Comment (AC1) · 30 Oct 2017

Reviewer 1: The analysis is solid; no problems. I would have liked to see some discussion about the variations among selected geographic regions or bio-provinces, but understand that the static description of those regions makes it difficult. Is it possible to run simple statistics to look for significant spatial differences? I admit I am not sure how powerful they would be, but is the reader to conclude that differences among the broad regions are insignificant? Perhaps a more strongly worded conclusion on that result would be helpful.

Authors Response: Given the number of biogeographical provinces represented in

this dataset we analysed the differences in the P-E parameters seasonally between adjacent provinces, using the Bonferroni post-hoc t-test. The results are summarized in a new figure (Figure 7) as well as in the text.

Reviewer 1: The authors also discuss the effects of phytoplankton composition on the parameter estimates. I wonder if using a broad description of pigments as available in ESSD might provide some insights into the variations observed in P-E parameters?

Authors Response: We agree that large-scale changes in phytoplankton composition present in the MAREDAT global database of HPLC pigment measurements published in ESSD (https://doi.org/10.5194/essd-5-109-2013) may provide some insight into latitudinal trends in community structure. We would argue however that the differences in the geographical distribution of the two datasets makes such comparisons difficult. We hope in the future that we may be able to add information on pigment markers to this database for experiments that have collected ancillary HPLC pigment samples.

L. Krug: This study offers a comprehensive and unprecedented global database of photosynthesis-irradiance parameters derived from more than 5000 experiments. The database encompasses quality-controlled parameters of different oceanographic regimes providing a global scale insight of phytoplankton photophysiological variation. In my opinion, it is very well written and illustrated and it is ready for publication. I would suggest a consistence in tables' format.

Authors Response: We formatted all Tables with identical borders/font type.

L. Krug: Also, in pag 8, line 3, I believe the sentence refers to table 5 instead of table 2

Authors Response: The table number was changed.

L. Krug: and in page 9 (line 14-16) you could refer to Figure 9.

Authors Response: The reference to Figure 9 was added.

A: Poulton: P1, Ln 39: Concentration of chlorophyll – please specify chlorophyll-a, or

are some of the normalisation against total chlorophyll? I assume the former is relevant based on reading the introduction and methods. As chlorophyll-a from both fluorometric and HPLC measurements are used for normalisation, should this not be another flag or comment in the database? Alternatively, the authors should add some comment on the sensitivity of derived parameters from fluorometric versus HPLC measures of chlorophyll biomass.

Authors Response: We agree that information on potential bias will be of interest to the reader and have included text on differences between fluorometric versus HPLC measures of chlorophyll biomass and its impact on the magnitude of the photosynthetic parameters.

A. Poulton: P2, Ln 4: Replace 'would' with 'will'.

Authors Response: Change has been made.

A. Poulton: P6, Ln 18-19: More rationale is needed to explain why the data from sea-ice algae with chlorophyll-a concentrations greater than 50 mg chl-a m-2 were removed. I can see why but a report detailing the construction of a database needs to establish clear rationale for data-removal (especially given that such a database may be expanded in the future).

Authors Response: There were a number of reasons for excluding these data from the database. First, the database focuses on marine phytoplankton. It is well-established differences in the ecological and physiological characteristics of sea-ice algae and marine phytoplankton necessitate that these two groups be treated separately. Moreover, the methodological differences of how the photosynthesis-irradiance experiments were conducted also meant that the datasets are not readily comparable. However, we hope that in the future an independent database for photosynthetic characteristics of sea-ice algae can be assembled when more experimental data becomes available.

A. Poulton: P6, Ln 23 onwards: Here it would be good to make clear that the application

of the Longhurst (1995, 1998) provinces is the author's choice to represent the eco-geographical spread of the data and is not inherent in the database.

Authors Response: We have clarified this in the text: that the assignment of provinces was based on geographic location of the data.

A. Poulton: P7, Ln 25-26: Please make it clear that chlorophyll-a concentrations have been used to classify trophic conditions (i.e. it is unclear (to the uninitiated) what 0.02 mg m-3 and 39.8 mg m-3 refer to, and there are other index's that could be used to examine trophic conditions).

Authors Response: We have changed the text to make it clear that chlorophyll-a concentration is our index of trophic status.

A. Poulton: P7, Ln 28-29: How are the data distributed? Have the authors considered using a geo- metric mean rather than an arithmetic mean? Rates of photosynthesis and chlorophyll- a concentrations range from very low values (e.g., oligotrophic waters) up to extremely high values (e.g., ice-edge bloom). Consequently, photosynthetic rates (and derived parameters) can vary over several orders of magnitude, and appear to exhibit a log- normal distribution (e.g. Fig 6). In this case maybe a geometric mean, rather than arithmetic mean, may better represent the data?

Authors Response: It should be noted that by normalizing the photosynthetic rates to chlorophyll biomass reduces the overall range of photosynthetic rate. Nevertheless, we have examined whether the geometric mean rather than arithmetic mean differs significantly. As the differences are small (averaging around 10-15%), we have decided to keep to reporting the arithmetic mean and standard deviation which is commonly reported in other studies.

A. Poulton: P11, Ln 18-19: What about nutrient availability as an environmental factor which varies strongly with latitude?

Authors Response: Variability of nutrients with latitude is complex, but we would agree

that in the central basins there may be a relationship, in particular between temperate and subtropical latitudes. We have added some text that mentions how latitudinal changes in heating and upper ocean stratification may also play a role in governing large-scale patterns of nutrient supply and hence ocean productivity.

A. Poulton: Figs. 3-8: Please ensure that the axis labels and data-points are clear in the final publication sized version of the manuscript to ensure they can be clearly read (some of the early figures have slightly small text for the axis).

Authors Response: We have increased the size of the fonts and symbols.

A. Poulton: Fig. 8 and 9: The density (or heat plot) component of the plots is not mentioned in the figure legends.

Authors Response: We have mentioned the density plot in the figure legend.

[Figure]

**Fig. 1.** Figure 7: Diagram illustrating the seasonal differences of the PE parameters between adjacent provinces by pairwise comparisons using Bonferroni adjusted t-tests.